# The Clinical Significance of CD73 in Cancer

**DOI:** 10.3390/ijms241411759

**Published:** 2023-07-21

**Authors:** Niklas Bach, Riekje Winzer, Eva Tolosa, Walter Fiedler, Franziska Brauneck

**Affiliations:** 1Department of Immunology, University Medical Center Hamburg-Eppendorf, 20246 Hamburg, Germanyr.winzer@uke.de (R.W.); e.tolosa@uke.de (E.T.); 2Department of Oncology, Hematology and Bone Marrow Transplantation with Section Pneumology, Hubertus Wald University Cancer Center, University Medical Center Hamburg-Eppendorf, 20246 Hamburg, Germany; f.brauneck@uke.de; 3Mildred Scheel Cancer Career Center HaTriCS4, University Medical Center Hamburg-Eppendorf, 20246 Hamburg, Germany

**Keywords:** CD73, immunotherapeutic strategies, immune suppression, cancer

## Abstract

The search for new and effective treatment targets for cancer immunotherapy is an ongoing challenge. Alongside the more established inhibitory immune checkpoints, a novel potential target is CD73. As one of the key enzymes in the purinergic signalling pathway CD73 is responsible for the generation of immune suppressive adenosine. The expression of CD73 is higher in tumours than in the corresponding healthy tissues and associated with a poor prognosis. CD73, mainly by the production of adenosine, is critical in the suppression of an adequate anti-tumour immune response, but also in promoting cancer cell proliferation, tumour growth, angiogenesis, and metastasis. The upregulation of CD73 and generation of adenosine by tumour or tumour-associated immune cells is a common resistance mechanism to many cancer treatments such as chemotherapy, radiotherapy, targeted therapy, and immunotherapy. Therefore, the inhibition of CD73 represents a new and promising approach to increase therapy efficacy. Several CD73 inhibitors have already been developed and successfully demonstrated anti-cancer activity in preclinical studies. Currently, clinical studies evaluate CD73 inhibitors in different therapy combinations and tumour entities. The initial results suggest that inhibiting CD73 could be an effective option to augment anti-cancer immunotherapeutic strategies. This review provides an overview of the rationale behind the CD73 inhibition in different treatment combinations and the role of CD73 as a prognostic marker.

## 1. Introduction

Immunotherapy marks a breakthrough in cancer therapy and has achieved impressive therapeutic effects [1]. Still, only 20–40% of patients respond, and the therapeutic effects are often short-lived due to the tumour’s high capacity to adapt and to develop resistance mechanisms [2]. While combinations of different treatments and further checkpoint targets are under investigation, a deeper understanding of the tumour microenvironment (TME) is crucial for the development of new therapies.

One component of the TME is the immunosuppressive metabolite adenosine. Tumours themselves and cancer therapies trigger the release of high amounts of ATP into extracellular compartments [3], which is then dephosphorylated by the ectonucleotidases CD39 and CD73 to produce adenosine [4]. The extracellular concentration of adenosine underlies a tight spatiotemporal control since adenosine is rapidly removed from the extracellular space by further degradation to inosine (catalysed by adenosine deaminase, ADA), by uptake into cells via concentrative or equilibrative nucleoside transporters [5]. The massive ATP release in the TME and its continuous degradation by the ectoenzymes CD39 and CD73 results in sustained high concentrations of adenosine [6], which engage A2A or A2B adenosine receptors on T cells, leading to an increase in cAMP that results in global suppression of T cell functions [7,8,9,10,11,12]. Dendritic cells stimulated with adenosine develop a pro-tumourigenic phenotype by expressing immune suppressor receptors and secreting angiogenic and tolerogenic factors [13]. Moreover, adenosine inhibits natural killer cell maturation and cytotoxic function [14,15], and modulates the function of other immune cells such as regulatory T cells (Tregs), macrophages, and neutrophils [16] (Figure 1). Besides suppressing the anti-tumour immune response, adenosine plays a role in multiple cancer-associated effects. High extracellular adenosine concentrations promote cancer cell proliferation/tumour growth, cell migration, angiogenesis, epithelial-mesenchymal transition, and metastasis [16,17,18]. Due to the major impact of purinergic mediators in the TME, the CD39/CD73/adenosine/A2AR pathway is emerging as a promising therapeutic target [18,19,20].

Here, we discuss the importance of CD73 as target in human cancers and elaborate current strategies for CD73 inhibition to reverse the immunosuppressive effects mediated by CD73-dependent adenosine generation.

## 2. Results

### 2.1. Biology and Regulation of CD73

The 5′-ectonucleotidase CD73 is a GPI-anchored glycoprotein located on the extracellular side of the plasma membrane [4,21]. The functionally active CD73 is a dimer of two identical subunits linked by non-covalent bonds. The N-terminal domain of CD73 contains two metal ion binding sites and is responsible for the phosphohydrolase activity. The C-terminal domain is responsible for the binding of the substrate AMP. The connection between these domains allows movement and thereby the switch between an open and closed conformation [22].

CD73 catalyses the hydrolysis of ribo- and deoxyribonucleosides 5′-monophosphates, most effectively the hydrolysis of AMP to adenosine with Km values of 1 to 50 µM [4]. ATP and ADP are competitive inhibitors of CD73 which bind to the catalytic site but cannot be hydrolysed [4]. Hence, the structures of ATP and ADP are often used as lead structures for the development of CD73 inhibitors. Besides its catalytic function, CD73 may play a role in cell adhesion and as a receptor for extracellular matrix proteins [4,23]. CD73 is shed from the cell membrane by cleavage of the GPI-anchor through hydrolysis or by phospholipases, generating soluble CD73. CD73 is also released on extracellular vesicles, for example from T cells upon activation [24,25] or from cancer cells [26,27,28].

A complex network of transcription factors regulates the expression of *NT5E*, the gene encoding CD73. The *NT5E* promotor contains binding sites for the transcription factors HIF-1α, AP2, SP1, Gfi1, different STAT and SMAD proteins, and cAMP responsive elements [29]. Relevant for the TME, hypoxia is a principal driver of CD73 upregulation by triggering the activation of the transcription factor HIF-1α [30,31]. Under inflammatory conditions, hypoxia-induced CD73 expression serves as a safety mechanism to protect the tissue against damage by adenosine-mediated suppression of the immune response [32]. In the hypoxic TME, tumour cells hijack this mechanism with beneficial effects on their survival and growth [33,34].

In addition, a number of soluble mediators are involved in the regulation of CD73 in the TME: Type I interferons, abundantly produced by tumour cells, enhance CD73 expression in endothelial cells [35,36]. TGF-β, important for the regulation of stemness and immune tolerance, upregulates CD73 in murine CD4^+^ and CD8^+^ T cells, DCs, and macrophages [37]. On MDSC, TGF-β induces CD73 expression by phosphorylating mTOR and HIF-1α activation [38]. This regulation by TGF-β creates a feedback loop with adenosine signalling because adenosine signalling induces the production and secretion of TGF-β, which in turn maintains CD73 expression [39]. In prostate cancer, tumour cell-derived exosomes containing PGE2 increase CD73 expression on dendritic cells, leading to an inhibition of the anti-cancer response [40]. In addition, in breast cancer, oestrogen and oestradiol signalling has a negative effect on the expression of CD73 [41].

Beyond transcriptional regulation, the expression of *NT5E* is epigenetically regulated. In malignant melanoma, higher CpG methylation is associated with lower CD73 expression and correlates with a better prognosis [42]. Moreover, an alternatively spliced and inactive shorter variant of CD73 is upregulated in hepatocellular carcinoma. This short CD73 variant forms an intracellular complex with the long version of CD73, leading to the proteasome-dependent degradation of the active CD73 [43].

In summary, the correlation of CD73 expression with HIF-1α, TGFβ, and PGE2 highlights the importance of the ectoenzyme as a biomarker of tumour development and progression. HIF-1α [44], TGFβ [45], and PGE2 [46] have themselves been described as regulators of progression in a variety of tumours.

### 2.2. Expression of CD73 in Human Cancer

Gene expression data from a large cancer patient cohort show that CD73 mRNA is upregulated in the majority of human solid cancers in comparison to matched normal tissues [47]. However, the expression of CD73 is heterogeneous across different human cancer types. The expression of CD73 is high in glioblastoma, thyroid carcinoma, sarcoma, pancreatic carcinoma, stomach adenocarcinoma, colorectal carcinoma, renal cell carcinoma, oesophageal carcinoma, thymoma, rectum adenocarcinoma, lung adenocarcinoma, non-small cell lung cancer, and acute myeloid leukaemia. Lower expression of CD73 is observed in cancers of the genitourinary system, such as endometrial, cervical, ovarian, uterine, prostate, and urothelial carcinoma, and in melanoma, breast cancer, and cholangiocarcinoma [42,48,49,50,51,52,53].

The expression of CD73 in non-tumour cells of the TME can also contribute to a tumour-promoting environment through enhanced activity of the purinergic pathway. Among the cell types expressing CD73 and other enzymes of the purinergic pathway in the TME are tumour-associated macrophages, B cells, myeloid-derived suppressor cells, and cancer-associated fibroblasts [54]. The expression of CD73 on cancer-associated fibroblasts is enhanced by adenosine receptor A2B signalling, in a feedforward circuit triggered by adenosine production concomitant to tumour cell death [55]. Not only cell-bound CD73 contributes to the adenosine generation in the TME, but also exosomes from different cancer types express CD73 and CD39 and show a high capacity of adenosine generation [27].

### 2.3. Relevance of CD73 as Target in Cancer Therapy

CD73 is involved in immune suppression, tumour growth, metastasis, angiogenesis, and drug resistance [56], therefore constituting a promising therapy target [57]. Tumour and host CD73 promote tumour growth by limiting T cell infiltration and activation [54], especially impairing the cytotoxic function of tumour-infiltrating CD8^+^ T cells [58]. CD73 can also affect the function of other immune cells, i.e., by promoting the differentiation of myeloid cells to tumour growth-promoting M2 macrophages [59]. The involvement of CD73 in the inhibition of an anti-tumour immune response has been confirmed in vivo in animal models: adoptive T cell immunotherapy cured all CD73 deficient mice, while it had no effect in mice with normal CD73 expression [60]. Further, fibrosarcoma and prostate tumours did not grow in CD73 deficient mice due to a strong anti-tumour response mediated by NK and CD8^+^ T cells [61].

CD73 directly regulates the tumour development. In vitro, CD73-mediated production of adenosine has shown to stimulate the proliferation of glioma cells [62] and inhibits apoptosis of ovarian tumour cells by upregulating the anti-apoptotic protein Bcl-2 [63]. In addition, CD73 is involved in cancer progression and metastasis by regulating the cell cycle via PI3K/AKT signalling [64,65]. Moreover, CD73 regulates cancer cell metabolism by promoting the Warburg effect. Genetic silencing and pharmacological inhibition of CD73 reduced glycolysis and cell proliferation in cancer cells, while overexpression promoted them [66]. An important step in metastasis is the epithelial–mesenchymal transition, and CD73 promotes the expression of stemness and EMT-associated genes [67,68,69]. Metastasis-conditioning features such as invasion, migration, and adhesion to the extracellular matrix were increased in CD73 overexpressing cell lines of human breast cancer, and these properties could be inhibited by blocking CD73 catalytic function [70]. Angiogenesis, another cancer-associated process, is also promoted by high CD73 expression. CD73-expressing endothelial cells formed more capillary-like structures than CD73-negative endothelial cells when cultured in cancer-conditioned medium [71]. Finally, the total ablation of CD73 in mice or its blockade in vivo using selective inhibitors reduced tumour growth and metastasis in multiple solid cancers models [72,73].

Furthermore, CD73 expression and purinergic signalling induce resistance mechanisms to anti-cancer therapy [74]. Mechanisms and implications for treatment will be discussed later in this review.

The increased expression of CD73 in tumour tissue and its role in the regulation of tumour cell proliferation, metastasis, metabolism, and angiogenesis, as well as inhibition of apoptosis, provide the rationale that blockade of CD73 may be of therapeutic benefit in the fight against cancer.

### 2.4. Prognostic Value of CD73

Considering the involvement of CD73 in the regulation of cancer-associated processes including proliferation, adhesion, migration, metastasis, and inhibition of the anti-tumour response [57], several studies investigated the correlation of CD73 expression with the clinical outcome of cancer patients. Overall, high CD73 expressions in tumour and host tissues predominantly showed negative prognostic value by promoting disease progression and metastasis [75]. However, studies also revealed that the prognostic value of CD73 differs in different tumour types and depending on the CD73-expressing cells.

To reveal the prognostic effect of CD73, gene expression data and immunohistochemistry were correlated with clinicopathological data. In breast cancer, especially triple negative breast cancer, a high CD73 expression is associated with a reduced overall and disease-free survival [76,77,78]. In contrast, a study conducted in a smaller patient cohort with breast cancer described a correlation of CD73 with longer disease-free survival [79]. Similar conflicting results were reported for ovarian cancer [63,80]. This may be due to small study cohorts and confounding expressions of other markers. Other studies found CD73 to be a marker of poor prognosis in gastric carcinoma [81], gallbladder carcinoma [68], cholangiocarcinoma [82], colorectal cancer [83], hepatocellular carcinoma [64], non-small cell lung cancer [84], papillary thyroid cancer [52], prostate cancer [85], and melanoma [86]. In these studies, CD73 expression correlates with worse grading, bigger tumour size, higher invasiveness, positive lymph node status, and increased metastasis. The opposite effects were observed in a cohort of patients with non-muscle invasive urothelial bladder cancer. Here, CD73 was associated with a lower stage, lower grade, and reduced proliferation [87]. In other cancers, for example acute lymphoid leukaemia, no association of CD73 with disease progression could be shown [88]. A review including the meta-analysis of 14 publications confirmed these heterogenic findings. CD73 expression was associated with a reduced overall survival in breast and ovarian cancer while positive effects were reported for lung and gastric cancer [47]. Further analysis of large datasets confirmed that high CD73 expression correlates with poorer clinical prognosis in most cancers. However, in some exceptions, such as endometrial carcinoma or clear cell renal carcinoma, CD73 appeared to be a protective factor [49,50]. Interestingly, the CD73 expression correlated with the expression of other immune checkpoints such as PD-L1 and could be used as a predictive marker for the efficacy of immunotherapy [49,50].

CD73 is investigated as a potential biomarker indicating the magnitude of treatment response in various cancer types. In a prospective cohort study it was shown that AMP hydrolysis in the blood plasma of elderly breast cancer patients was higher than in healthy elderly women and significantly decreased after every mode of therapy [89]. The measurement of CD73 and adenosine signalling could be useful to select patients for therapies that modulate the purinergic pathway. As it is not possible to measure extracellular adenosine levels in the clinical routine, other markers and specific gene signatures of adenosine signalling are investigated as biomarkers [90].

In addition to membrane-bound CD73, active CD73 also exists as a homodimer in a soluble form [24]. The expression and activity of soluble CD73 in serum was retrospectively analysed in a multicentre study as biomarker in patients with metastatic melanoma. Melanoma patients showed higher CD73 activity and expression than healthy donors. In addition, elevated CD73 activity levels before and during treatment were associated with nonresponse to therapy with nivolumab or pembrolizumab. Multivariate Cox regression analyses showed that serum CD73 was an independent prognostic factor for both overall survival and progression-free survival [91]. Messaoudi et al. [92] analysed the prognostic relevance of soluble CD73 and intratumoural CD73 in patients with resected liver metastases from colorectal cancer. High intratumoural CD73 concentration was associated with shorter disease-specific survival and time to recurrence, multiple and larger metastases, and resistance to preoperative chemotherapy. Interestingly, soluble CD73 did not correlate with intratumoural CD73. A correlation with reduced disease-specific survival was only observed for a subset of patients with highest soluble CD73 levels (7.2%) [92].

Cancer cells as well as activated CD8^+^ T cells secrete small vesicles, so-called extracellular vesicles (EVs)s, which contain active CD73. Extracellular adenosine production also takes place by CD73 present on the EVs, and this can indirectly modulate the TME. EVs are increasingly investigated and used as a new reservoir for cancer biomarker discovery [24,25,27]. In a retrospective pilot study, exosomal CD73 was assessed before and at the start of treatment with anti-PD-1 agents in patients with melanoma. At baseline, CD73 and PD-L1 expression levels on EVs derived from patients receiving pembrolizumab or nivolumab monotherapy were significantly increased. In addition, CD73^+^ EVs increased significantly during treatment in patients who did not respond to therapy [93].

Taken together, the multiple correlations of CD73 with disease prognosis, with therapy response, or with cancer-associated regulators such as HIF-1α, TGFβ, or PGE2 underscore its role as a biomarker. Because the analysis of membrane-bound CD73 from TME is invasive, it seems promising to analyse soluble CD73 or CD73^+^ exosomes as biomarkers. Further studies are needed to evaluate these new diagnostic tools for the clinic.

### 2.5. CD73 Inhibition in Cancer

CD73 represents an ideal target in cancer therapy. Overexpression of CD73 on tumour and immune cells leads to increased adenosine concentration in the TME, which inhibits antitumour immune responses via different mechanisms and promotes proliferation, angiogenesis, and metastasis. CD73 inhibition or deficiency shows impressive anti-tumour effects in preclinical experiments with mice, while it is associated with only mild adverse events [57]. In humans, loss-of-function mutations in *NT5E* cause calcification of peripheral arteries and a higher risk of cardiovascular disease [93,94].

Clinical phase 1 and 2 trials with different CD73 inhibitors are ongoing for advanced and metastatic cancers [95]. Few of them have been completed, but published results give reasons to be optimistic. CD73-targeting therapies have shown to be well-tolerable and clinically active [57]. In particular the combination of CD73 inhibitors with other cancer treatments seems to be promising [19,96], as CD73 is involved in the development of various treatment resistances against chemotherapy, radiotherapy, target therapy, and immunotherapy [48] (Figure 2). The different treatment combinations and underlying rationale will be discussed in the following sections.

#### 2.5.1. Drugs for CD73 Inhibition

In the past years, a plethora of CD73 inhibiting agents has been developed. Many of them are currently being evaluated in clinical trials (Table 1). The two main groups of CD73 inhibitors are small molecule compounds and monoclonal antibodies. Both have several advantages and disadvantages. Advantages of small-molecule inhibitors are the easier and possible oral application and the enhanced distribution with a deeper tumour penetration, including the crossing of tissue barriers [97]. Disadvantages are the lower specificity and possible off-target effects caused by their binding to more conserved sites across various enzymes. In contrast, a higher specificity and lower toxicity are the biggest advantages of monoclonal antibodies, while high development costs and worse tissue penetration are disadvantageous [98]. Another important feature is the capacity of inhibitors to reach the intracellular and thereby to block also intracellular CD73 [99].

The first small-molecule CD73 inhibitor and starting point for further developments was adenosine 5′-(α,β-methylene)diphosphate (APCP), a non-hydrolysable ADP analogue. Since then, much work has been done in designing more stable derivatives of APCP [97]. Great advances have been made since the X-ray co-crystal structure of CD73 in complex with APCP was reported. This allowed targeted modifications of the nucleobase, sugar, and zinc-binding groups and led to the development of many highly effective and promising inhibitors [100]. All of the originating compounds are so-called nucleotide-based small-molecule and competitive inhibitors, since they directly bind to the catalytic site [101]. One promising candidate is AB680, a highly potent and selective CD73 inhibitor with improved metabolic stability [97]. In preclinical experiments it restores the adenosine-mediated inhibition of anti-tumour immune responses [102,103]. It is also the first small-molecule CD73 inhibitor tested in ongoing clinical trials. Initial clinical data supports a good toleration with few and mild adverse events and a good pharmacokinetic profile [104]. Meanwhile, two more small-molecule inhibitors, ORIC-533 and LY3475070, are investigated in clinical phase 1 studies [97]. Besides the nucleotide-based inhibitors, a growing number of non-nucleotide small molecule inhibitors has been developed. The most promising chemical structural families for the development of this kind of inhibitors are sulphonamides, anthraquinones, and other flavonoids [105]. In order to improve the specificity, allosteric binding small-molecule inhibitors are of interest [101]. In this context, the dimerization interface of CD73 has been reported to be a promising target site [106].Many monoclonal antibodies targeting CD73 have been developed and proven to cause anti-cancer effects in preclinical experiments [107,108,109]. Oleclumab (MEDI9447, AstraZeneca, Gaithersburg, MD, USA) was the first monoclonal antibody to enter clinical trials [110], followed by various others (Table 1). Meanwhile, initial clinical data has been collected, supporting a good safety and tolerability of the investigated antibodies [111,112]. Differences between the antibodies, for example a varying strength of the “hook effect” or incomplete enzymatic inhibition, are mainly due to different binding epitopes [113]. Oleclumab has a dual mechanism of action. It causes the crosslinking of CD73 dimers and blocks CD73 from adopting its catalytically active form [114]. Other antibodies directly bind to the catalytic site [113]. However, the inhibition of CD73’s enzymatic activity is only one of the mechanisms contributing to the effects of CD73 blockade. Other studies reported that binding of antibodies can trigger clustering and internalisation of membrane-bound CD73. This inhibits extravasation and colonisation of tumour cells and therefore reduces the formation of metastasis [115]. Another mechanism of action is the crosslinking of Fc receptors that triggers host mechanisms such as the antibody-dependent cellular toxicity or complement activation [116]. The development of monoclonal antibodies allows the design of highly individual binding regions and constructs. One approach is to design bispecific antibodies, which can recognise more than one epitope with two different Fab arms. This was reported to cause a greater inhibition of CD73 due to additional effects [117]. Another anti-CD73 antibody is fused to the extracellular domain of the TGF-β receptor II. This allows the additional trapping of immunosuppressive TGF-β [118].

The development of nanobodies has brought some interesting possibilities as these biologics combine the high specificity and low toxicity of antibodies to the better tissue penetration of small-molecule inhibitors [98]. Especially constructs that bispecifically target CD73 and PD-L1 cause anti-tumour effects in mouse melanoma models [119].

Drug repurposing is a cost-effective alternative to find inhibitors of CD73. In the screening of hundreds of compounds, several promising candidates were found. Dasatinib, a tyrosine kinase inhibitor targeting BCR-ABL in CML, inhibits CD73 and modulates the TME. It also blocks TGF-β-induced expression of EMT-promoting transcription factors. Another promising compound is Pentoxifylline (Sanofi, Paris, France), originally approved for the treatment of peripheral vascular diseases. The combination of Dasatinib (Bristol Myers Squibb, New York City, NY, USA) and Pentoxifylline is described as a possible treatment option in cancer [48].

#### 2.5.2. Monotherapy

CD73 blockade causes huge anti-tumour effects in preclinical experiments [20]. This is based on a better immune response, including enhanced NK cell activity and CD4^+^ and CD8^+^ T cell function and increased levels of proinflammatory cytokines, mainly IFN-γ [120,121]. Besides the enhanced immune response, also other effects have been observed. The inhibition or downregulation of CD73 causes a decrease in tumour VEGF levels impairing tumour angiogenesis [122,123]. In addition, CD73 inhibition decreased the capability for autophagy of tumour cells [124] and induced apoptosis and cell cycle arrest [125]. Another described mechanism was the reduction in tumour metastasis upon antibody binding induced clustering and internalisation of CD73 [115].

Based on the preclinical data, several phase I trials are currently ongoing, testing monoclonal antibodies as well as small molecule inhibitors directed against CD73 for advanced solid tumours including non-small cell lung cancer, triple negative breast cancer, pancreatic ductal adenocarcinoma, colorectal cancer, renal cell carcinoma, and prostate cancer, as well as refractory multiple myeloma (NCT04148937, NCT05246995, NCT05173792, NCT04672434, NCT05143970, NCT05431270, NCT05227144) [108,126,127,128]. The majority of the studies are currently recruiting, and no results have been announced yet. The antibody Oleclumab is furthest in clinical testing. Monotherapy showed a manageable safety profile for advanced solid malignancies with no dose-limiting toxicities and an exposure profile consistent with those of other monoclonal antibodies. Evidence of anti-tumour activity was observed, particularly in tumour types that are generally resistant to immunotherapy [96]. Overall, the monotherapy with Oleclumab had only small clinical efficacy so far [96,112,129] with only marginally improved overall response rate (ORR) [130]. Given the mechanistic rationale and the successful preclinical data, further studies are investigating the combination of Oleclumab plus Durvalumab or with standard of care chemotherapies.

#### 2.5.3. Combination with Other Inhibitors of the Purinergic Pathway

Sole enzymatic blockade of CD73 may not be sufficient to fully prevent adenosine generation, as alternative pathways for adenosine generation exist. It seems rational to additionally block other purinergic enzymes or receptors. The most studied combination partners of CD73 inhibition in cancer models are CD39 and the A2AR. For both exist multiple inhibitors that are currently being tested in clinical trials [19]. An advantage of targeting adenosine receptors themselves is, that all adenosine-mediated effects are inhibited, independently of the origin of adenosine [99].

CD39 and CD73 are both part of the main degradation pathway of ATP to adenosine. It was shown that the expression of both is associated with worse survival in cancer and that targeting both together causes better anti-tumour activity than just CD39 or CD73 alone [59]. As CD39 degrades ATP to ADP and AMP, its inhibition has a double effect by also increasing the level of ATP which stimulates the immune system, especially in combination with therapies leading to a high ATP release such as chemotherapy or radiotherapy [131]. The first phase I combination study with an CD39 inhibitor (IPH5201), which is used as monotherapy or in combination with Durvalumab ± Oleclumab in patients with advanced solid tumours, has just been completed. The results remain to be seen (NCT04261075).

Regarding the combinatorial targeting of CD73 and A2AR, it has been shown that in mice lacking CD73 and A2AR, a synergistic effect on tumour control and metastasis was observed in association with higher tumour infiltration by CD8^+^ T cells [132]. It was also described that the loss of A2AR signalling increases CD73 expression. This feedback loop is interrupted by the dual inhibition of CD73 and A2AR [133]. Several clinical trials currently assess the safety and efficacy of treatment combinations targeting the purinergic pathway in advanced cancers (NCT03454451, NCT03549000). In first results the targeting of CD73 by the monoclonal antibody NZV930 in combination with PD-1 and A2AR inhibition showed a tolerable safety with frequent but mild adverse events and no dose limiting toxicities. Although a decrease in adenosine in plasma and tumours has been measured, no objective responses to the treatment was observed and only 11% of the patients had a stable disease [127].

In addition, the effects of other adenosine receptors have been studied. A2BR signalling promotes the upregulation of CD73 on cancer-associated fibroblasts. This is blocked by A2BR inhibition, leading to an enhanced anti-tumour activity [55]. In vitro, agonistic targeting of the A3R was found to increase T cell activation in combination with CD73 and PD-1 inhibition [134]. This again underlines the complex and non-redundant effects of the purinergic pathway.

In summary, the adenosine pathway is a well-characterized mediator of immunosuppression in the TME, but the sequentially combined blockade of this pathway has shown limited clinical success so far. Based on the preclinical data, it is plausible that a strong reduction in extracellular adenosine in tumour patients by a blockade as complete as possible will lead to a significantly reduced immune suppression. However, the results of the current studies remain to be seen.

#### 2.5.4. Combination with Immunotherapy

Many cancer patients develop resistance to immune checkpoint inhibitors during treatment, even if the therapy has initially been effective [99]. This resistance is often associated with an upregulation of CD73 in the TME. For example, in human melanoma, a subset of patients with progressive disease and dedifferentiated tumours during immunotherapy, was marked by high CD73 expression [48]. A gene expression signature of adenosine signalling was associated with reduced efficacy of anti-PD-1 treatment in published cancer cohorts [90]. Interestingly, a high basal level of soluble CD73 could predict a worse response to immune checkpoint therapy with the PD-1 blocker nivolumab [86]. In melanoma patients, the expression of CD73 on exosomes was higher in patients that did not respond to anti-PD-1 therapy [135]. A population of persisting CD73^hi^ myeloid cells after anti-PD-1 treatment was associated with a reduced overall survival in a cohort of patient with glioblastoma multiforme. In following preclinical experiments using CD73^-/-^ mice treated with anti-CTLA-4 or anti-PD-1 the survival was improved, identifying CD73 as a combinatorial treatment target [136].

The synergistic effect of CD73 inhibition in combination with immune checkpoint inhibitors (ICI) was confirmed by other preclinical studies [75]. The combination of CD73 inhibition and anti-CTLA-4 or anti-PD-1 treatment enhances the anti-tumour immune response and inhibits tumour growth [134,137]. This was mediated by a restored immune functionality [102], including enhanced CD8^+^ T cell function and increased production of IFN-γ and granzyme-B by tumour-infiltrating lymphocytes [131,138,139]. The upregulation of CD73 during immune checkpoint inhibition is an acquired resistance mechanism. This was confirmed in a breast cancer tumour model in mice. Anti-PD-1 treatment increased adenosine concentrations in the tumour tissue and this was suppressed by CD73 inhibition [140]. Mechanistically, the upregulation of CD73 in melanoma patients under immune checkpoint inhibition has been driven by mutations in the MAPK signalling pathways and TNF-α signalling [141]. In hepatocellular carcinoma, upon anti-PD-1 treatment, tumour-derived exosomes containing circular RNA upregulate also CD39 expression on macrophages [142]. Vice versa, CD73 and adenosine signalling are involved in the regulation of various immunosuppressors. Activation of A2AR enhances especially the expression of PD-1 on tumour-specific CD8^+^ T cells [143]. Adenosine l, via cAMP and PKA activation, leads to the inhibition of T cell receptor signalling and the activation of the transcription factor CREB, which promotes the expression of immunosuppressors such as PD-1 and CTLA-4 but also TGF-β and IL-10 [33]. Furthermore, adenosine signalling reduces the expression of cyclin-D1, promoting the expression of PD-L1. This effect was reversed by CD73 inhibition [144].

Due to the promising results of preclinical studies, the combination of CD73 inhibitors with immune checkpoint inhibitors is currently tested in clinical trials in patients with different advanced solid tumours. Most of these studies reported tolerable safety and preliminary clinical activity [127,145,146,147]. One trial (NCT03822351) evaluated the combination of CD73 inhibitor Oleclumab and Duvarlumab (anti-PD-L1) and was conducted with 189 patients with stage 3 NSCLC. Response rates were higher in the combination group than with anti-PD-L1 therapy alone (ORR 30% vs. 17.9%). Additionally, the progression-free survival (PFS) had almost doubled in the combination cohort (PFS after 12 months: 62.6% vs. 33.9%). No significant different in therapy safety were observed [126]. Based on these positive results, the first phase 3 trial with CD73 inhibition in combination with immune checkpoint inhibition was initiated (NCT05221840). Another clinical trial reported less-promising results. In a solid cancer study cohort, CD73 levels on tumour and T cells were decreased upon treatment with Oleclumab, accompanied by low anti-tumour activity and very low response rates [96].

Besides the combination of CD73 with immune checkpoint inhibitors, the combination with agonists of costimulatory molecules such as 4-1BB (CD137) or OX-40 was also preclinically tested and found to enhance anti-tumour T cell immunity [95,148]. Likewise, the efficacy of other cellular immune therapies such as adoptive T cell transfer and CAR NK cells were enhanced by additional CD73 blockade in mouse models [95].

The current clinical data are not yet sufficient to confirm the additional benefit of inhibiting CD73 in combination with classical checkpoint inhibition in patients with advanced cancer. A number of studies are ongoing where the results remain to be seen.

#### 2.5.5. Combination with Chemotherapy

CD73 upregulation and adenosine production is described as an acquired resistance mechanism of tumour cells to chemotherapy. In line with this hypothesis, increased expression of CD73 on tumour cells have been observed after chemotherapy [149,150]. Regarding haematological malignancies, the most upregulated gene in chemotherapy-resistant leukemic cells was found to be CD73, which is part of a multi-resistance program and protects against TRAIL-induced apoptosis. This resistance can be acquired by normal cells via transfection with CD73 or removed by CD73 downregulation [151]. GWAS studies identified CD73 as a major determinant involved in resistance to platin-based chemotherapies [152]. In a mouse model of triple-negative breast cancer, other immunosuppressors, CD47 and PD-L1, were also upregulated by a HIF-1α-dependent transcriptional mechanism upon chemotherapy [149]. Chemotherapeutic drugs trigger extensive cell damage and death and thereby lead to the release of ATP, which is degraded to immunosuppressive adenosine within the TME. CD73 inhibition can reverse this immunosuppressive effect [134]. In a mouse model of ovarian cancer, CD73 inhibition caused decreased tumour growth and metastasis in combination with the chemotherapeutic drug Doxetaxel [150]. A retrospective analysis of samples from rectal cancer patients revealed that purinergic signalling is enhanced after chemo- and radiotherapy. Besides promoting the adenosine-mediated immunosuppressive effects, CD73 can also directly regulate drug resistance in cancer. The multi-resistance protein 1 (MRP1) is upregulated in chemotherapy treated cervical cancer and glioblastoma cells, causing the resistance to various chemotherapeutic drugs. This mechanism was regulated by adenosine via the A3 receptor. CD73 inhibition decreases MRP1 expression and makes the cancer cells more sensitive to chemotherapy, associated with better treatment responses [153,154]. CD73 can also increase chemoresistance by regulating intracellular NAD^+^ levels. These play a role in DNA repair mechanism and thereby enhance resistance to chemotherapy induced DNA damage [48].

To our knowledge, there are currently no clinical trials investigating a combination with chemotherapies. However, preclinical data suggest that blockade of CD73 has the potential to reverse chemotherapeutic resistance mechanisms.

#### 2.5.6. Combination with Radiotherapy

Besides causing direct damage to cancer cells, radiotherapy triggers a systemic anti-tumour response mediating immunogenic cell death in tumours. An important driver of this immune activation is the radiotherapy-induced release of DAMPs such as ATP from the intracellular space. An enhanced ATP degradation and adenosine generation constitutes an adaptive resistance mechanism to radiotherapy [155]. Considering this, the inhibition of CD73 in combination with radiotherapy is rational [156].

In preclinical mouse models it was shown that the anti-tumour effect of radiotherapy depends on the host’s immune function. CD73 and adenosine levels are significantly increased after radiation therapy and the inhibition of CD73 has synergistic effects on decreasing tumour growth and metastasis in combination with radiotherapy. This is due to an increased immune function [157,158]. Similar effects were observed with the combination of radiotherapy and A2AR antagonists [159]. The proteomic analysis of irradiation-selected pancreatic cancer cells revealed that *NT5E* is one of the most upregulated genes in a network of growth factors and cytokines that mediates radio-resistance by enhancing DNA repair, inactivating the proapoptotic protein BAD and promoting epithelial mesenchymal transition. In pancreatic cancer cells, CD73 overexpression results in radio-resistance while the knockdown of CD73 resensitises cancer cells to radiotherapy [160]. Moreover, higher CD73 activity and adenosine levels after irradiation of the thorax in mice were found to promote lung fibrosis, a major adverse side effect. In CD73 deficient mice this radiation induced lung fibrosis is milder [161].

A currently ongoing phase 2 clinical trial (NCT03875573) tests the effect of stereotactic body irradiation in combination with CD73 inhibition, anti-PD-L1 treatment, and neoadjuvant chemotherapy in luminal B breast cancer. The authors propose that this combination supports the activation of innate immune cells and induces T cell priming. This may turn the primary tumour into an individual tumour vaccine and thereby help to gain tumour control and to prevent metastasis [162].

#### 2.5.7. Combination with Targeted Therapies

The expression of CD73 is intertwined with the regulation and mutation of other targetable proteins that promote tumour growth. For example, activating mutations in the MAPK promote the expression of CD73 [95]. CD73 overexpression regulates cancer growth via the EGFR/AKT1 pathway. The inhibition of EGFR and AKT1 inhibits the proliferation of CD73^+^ tumour cells [163]. In addition, in breast cancer, CD73 was described to promote EGFR expression, most likely via the transcription factor PPAR-γ. CD73 inhibition can reverse while the addition of adenosine is promoting this effect [164]. Additionally, a correlation with BRAF/MEK activity was described. In BRAF mutant melanoma, the inhibition of BRAF and MEK causes a downregulation of CD73 tumour cells. The combination with an A2AR antagonist improves the protection against tumour initiation and metastasis in a mouse model [165]. A phase 1 clinical trial (NCT03381274) was conducted in a patient cohort with advanced and EGFR mutated NSCLC and assessed the combination therapy with Oleclumab and Osimertinib, an inhibitor of the EGFR tyrosine kinase. Acceptable tolerability and moderate clinical activity have been found. The overall response rate was low (ORR 11.8%), but patients who responded had a doubled response duration in comparison to monotherapy with Osimertinib (16.6 month vs. 7.4 month) [166].

In a clinical trial evaluating the efficacy of the anti-HER-2 monoclonal antibody Trastuzumab, high CD73 expression was associated with a poorer clinical outcome. In a mouse model, CD73 expression significantly suppressed the response to anti-HER-2 therapy. The additional blockade of CD73 increased the treatment efficacy [77]. A further phase 1 study is ongoing evaluating the enzymatic blockade of CD73 alone versus a combination with chemotherapy and Trastuzumab in patients with advanced solid tumours (NCT05143970).

Another interesting approach is the dual blockade of CD73 and TGF-β with a bifunctional antibody construct (Dalutrafusp). Both factors are known to correlate with EMT, fibrotic stroma, immune tolerance, and poor prognosis in triple-negative breast cancer. Preclinical experiments in mice reported a superior activity of the bifunctional antibody construct compared to CD73 blockade alone. Cell migration, EMT, and metastasis were reduced, and tumour cell death was induced, and proinflammatory conditions were established in the TME [167]. A first in-human clinical trial (NCT03954704) was performed in patient cohort with advanced solid tumours. The treatment was well tolerated and plasma TGF-β levels became undetectable during therapy, whereas levels of soluble CD73 were increased in comparison to the baseline [118]. The overall response rate was 38.1%, providing the rationale for further clinical evaluation. Based on the early clinical and preclinical data the combination of CD73 inhibition with targeted therapy represents a promising approach to overcome therapeutic resistance in the TME.

## 3. Conclusions

CD73 contributes to the development of numerous cancer-specific hallmarks. Its presence on tumour cells and other cell types in the TME, but also in soluble form or EV-bound, is suppressing the hosts anti-tumour immune response and enhances tumour growth, angiogenesis, EMT, cancer cell invasion, and metastasis. The primary mechanism behind this suppression is through adenosine signalling, making the catalytic activity of CD73 crucial in this context. In addition, direct non-enzymatic effects of CD73 are involved, for instance in the regulation of cell adhesion. CD73 is upregulated in most tumour types and often associated with resistance to standard of care cancer therapies and a poor survival. Furthermore, CD73 is considered as a novel biomarker for some cancers.

Based on its overexpression on many tumours, CD73 might be considered as a pan-cancer biomarker. In most studies, a high CD73 expression was associated with a worse prognosis [49,50]. Recent studies suggest that CD73 may also serve as a biomarker for the prediction of immunotherapy responses [49,50]. Moreover, CD73 has been shown to be further upregulated during the development of immunotherapy resistance, especially in (platinum)-containing chemo- and radiotherapies [153,154,160]. Due to the invasiveness of determination of membrane-bound CD73 in tumour tissue, the analysis of soluble CD73 and CD73^+^ EVs from biological fluids represents a promising new approach for clinical practice.

The blockade of CD73 results in minimal adverse effects in patients. Mostly mild and unspecific treatment-related adverse events have been observed, including fatigue, nausea, vomiting, diarrhoea, headache, fever, elevated liver enzymes, or cough [96,118,127,145]. When combined with the PD-L1 blocker Durvalumab, no additional adverse events have been observed in comparison to the treatment with Durvalumab alone [126]. In rare cases the adverse event pneumonitis led to discontinuation of single-study participants [126,166].

Until now, only phase 1 and 2 clinical trials with few patients with advanced and highly heterogeneous tumours have been conducted, while preclinical studies in mice used models of non-advanced and homogeneous cancers. In addition, Phase I and II studies have the focus on safety. Therefore, no definitive statement on the efficacy can be made. Furthermore, the correlation between CD73 expression and efficacy remains to be investigated in larger patient cohorts. In addition, the role of CD73 expressing T cells, NK cells or tumour-associated macrophages has not yet been investigated in these studies. Future trials involving bigger patient cohorts should provide a more comprehensive understanding.

Preclinical studies have shown that CD73 and adenosine signalling are intertwined with the regulation and mutation of other targeted proteins, and with the promotion of resistance mechanisms of tumour cells to classical checkpoint blockade and to radio- and chemotherapeutic agents. Therefore, especially combinational treatment approaches are efficient in reducing tumour growth and metastasis. Any factor that contributes to the regulation of cancer-promoting properties within TME can be a suited combination partner for CD73 blockade. Undoubtedly, as our understanding of the TME improves, the number of potential treatment combinations will continue to expand. Especially the combination with immune checkpoint inhibitors is currently studied in ongoing clinical trials. Additionally, other players of the purinergic pathway represent promising targets in cancer therapy. To date, clinical trials have primarily focused on investigating inhibitors of CD39 and the A2AR, in addition to CD73. Selecting the most effective treatment combinations tailored to each individual patient is crucial for optimal outcomes. Therefore, meaningful biomarkers are needed.

It will be interesting to see the results of the many ongoing clinical trials using CD73 inhibition in combination with other treatments for cancer therapy. Hopefully, they will provide the basis for further phase 3 clinical trials, leading to the approval of CD73 inhibitors for clinical use—and therefore to the addition of a new and efficient treatment in cancer.

## Figures and Tables

**Figure 1 ijms-24-11759-f001:**
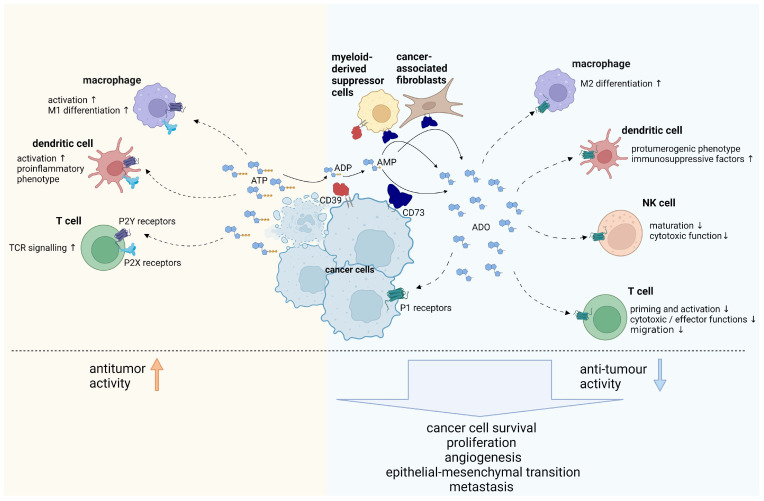
ATP and adenosine signalling in the tumour microenvironment (TME). ATP, released by damaged or dying cells, is a danger signal to immune cells and mainly promotes an anti-tumour immune response by signalling through P2X and P2Y receptors, which are differentially expressed across cell types. ATP is degraded to adenosine by the concerted action of CD39 and CD73. Adenosine promotes tumour growth by suppressing immune cells by way of A2A and A2B receptor signalling, or by directly enhancing tumour cell proliferation, angiogenesis, epithelial-mesenchymal transition (EMT), and metastasis. ADO: Adenosine. ↑: increased. ↓: decreased. Solid arrow: gets degraded/dephosphorylated. Dashed arrow: binds receptor and signals. Created with BioRender.com.

**Figure 2 ijms-24-11759-f002:**
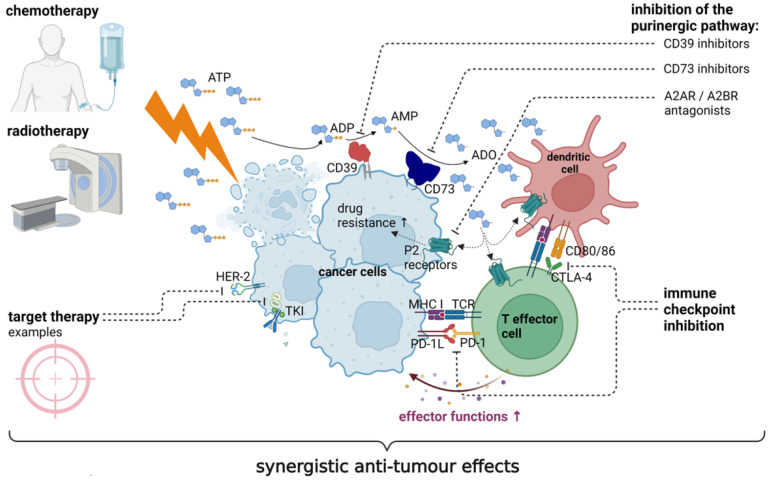
The synergistic anti-tumour effect of combined anti-cancer treatments. The efficacy of many treatments, such as chemotherapy, radiotherapy, target therapy, and immunotherapy relies on the activation of an anti-tumour immune response. The upregulation of CD73 is a resistance mechanism to anti-cancer therapies. While ATP, released from stressed and dying cells, supports the immune activation, adenosine, generated by CD73 catalytic activity, dampens the activity of immune cells, and triggers several drug resistance pathways in tumour cells. Combining blockers of the purinergic pathway (CD39 or CD73 inhibitors, A2AR or A2BR antagonists) or the combination of CD73 inhibition with other treatments (immune checkpoint inhibition, chemotherapy, radiotherapy, target therapy) causes synergistic anti-tumour effects, for example by promoting immune effector functions. ↑: increased. Solid arrow: gets degraded/dephosphorylated. Dashed arrow: inhibits. Created with BioRender.com.

**Table 1 ijms-24-11759-t001:** CD73 inhibitors that are currently evaluated in clinical trials. Trial information obtained from https://clinicaltrials.gov (accessed on 10 May 2023).

Type	Inhibitor	Developer	Trials
Small molecule inhibitor	AB680 (Quemliclustat)	Arcus Biosciences (Hayward, CA, USA)	NCT04104672, NCT04660812
	ATG037	Atengene (Shanghai, China)	NCT05205109
	LY3475070	Lilly Pharma (Indianapolis, IN, USA)	NCT04148937
	ORIC533	Oric Oharmaceuticals (San Francisco, CA, USA)	NCT05227144
Monoclonal antibody	MEDI9447 (Oleclumab)	Medimmune/ AstraZeneca (Gaithersburg, MD, USA)	NCT02503774
			NCT03267589
			NCT03381274
			NCT03616886
			NCT03875573
			NCT04262375
			NCT04262388
			NCT04668300
			NCT04940286
	BMS986179	Bristol-Myers Squibb (New York, NY, USA)	NCT02754141
	AKK119	Akeso (Guangzhou, China)	NCT04572152
			NCT05173792
			NCT05559541
			NCT05689853
	CPI006 (Mupadolimab)	Corvus Pharmaceuticals (Burlingame, CA, USA)	NCT03454451
	HLX23	Henlius (Shanghai, China)	NCT04797468
	IB325	Innovent Biologics (Suzhou, China)	NCT05119998
			NCT05246995
	INCA00186	Incyte Corporation (Wilmington, DE, USA)	NCT04989387
	IPH5301	Innate Pharma (Marseille, France)	NCT05143970
	JAB-BX102	Jacobio Pharma (Beijing, China)	NCT05174585
	NZV930	Novartis (Basel, Switzerland)	NCT03549000
	PT199	Phanes Therapeutics (San Diego, CA, USA)	NCT05431270
	Sym024	Symphogen (Lyngby, Denmark)	NCT04672434
	TJ004309 (Uliledlimab)	I-Mab Biopharma (Rockville, MD, USA)	NCT04322006
			NCT05001347
Bifunctional antibody construct	AGEN1423 (Dalutrafusp) (GS-1423)	Agenus Gilead (Lexington, MA, USA)	NCT03954704
			NCT05632328

## Data Availability

Not applicable.

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
