# Peer review of "The Clinical Significance of CD73 in Cancer"

_ijms, 2023, doi:10.3390/ijms241411759_

Round 1

Reviewer 1 Report

  1. I have a few recommendations aimed to enhance the clarity, depth, and impact of the review.

  2. 1.- The review mentions the contribution of CD73 to various cancer-specific hallmarks, but it would be beneficial to include specific examples or studies that support these claims. Providing specific details and references can strengthen the arguments presented.

  3. 2.- Clarify the significance of findings: While the review acknowledges the role of CD73 in tumor development and its potential as a therapeutic target, it would be helpful to clearly explain why these findings are essential. What implications do these findings have for cancer treatment or patient outcomes? Providing a clear and concise statement about the research's significance can enhance the review's impact.

  4. 3.- Discuss limitations and challenges: The review briefly mentions that clinical trials have not yielded results as impressive as expected based on preclinical studies. It would be valuable to elaborate on the possible reasons for this discrepancy. Are there challenges or limitations in translating preclinical findings to clinical settings? Discussing these aspects can provide a more comprehensive understanding of the field and highlight areas for future research.

  5. 4.- Address potential adverse effects and considerations: The review mentions that CD73 blockade results in minimal adverse effects, but it also acknowledges that CD73 is involved in various physiological processes. It would be important to discuss the potential adverse effects or disruptions that could arise from CD73 inhibition and how they could be mitigated. Addressing these considerations would provide a balanced perspective on using CD73 inhibitors as therapeutic agents.

  6. 5.- Emphasize the need for biomarkers: The review briefly mentions the need for meaningful biomarkers in cancer therapy. It would be helpful to expand on this point and discuss the importance of identifying biomarkers that can guide treatment selection and predict patient response. Exploring the current challenges and potential advancements in biomarker research can add depth to the discussion.

Reviewer 2 Report

The manuscript by Bach et al. represents a review article focusing on CD73 targeting for immunotherapy of cancer. The topic of this review is timely and important. However, un my opinion the manuscript should include a number of improvements prior to the publication.

- As manuscript title state the focus of this review is CD73 targeting. Nonetheless, the main body of the review contains parts (2.1-2.5) dedicated to adenosine signaling, CD73 biology, expression and prognostic value. Each of these parts looks very general, insufficiently detailed, does not correspond to the major topic of the review directly and, moreover, represents a topic for independent review. Therefore, I would like to suggest the authors restriction and incorporation of these parts into the Introduction.

- The review does not provide clearly the authors’ opinion on the following questions:

- For which cancer types CD73 targeting is the most promising?

- What are the benefits and/or restrictions of each particular drug?

- Similar question about their use in combination with other inhibitors.

- It is interesting to see in the conclusions the ideas of the authors on the most promising areas for the future development of this topic.

Minor

- I think that the results of the authors’ analysis could be illustrated better than within the current version.

- The fonts with in the figures could be enlarged.

Round 2

Reviewer 2 Report

the concerns have been addressed